# Development and Validation of a Meta-Instrument for Nursing Assessment in Adult Hospitalization Units (VALENF Instrument) (Part I)

**DOI:** 10.3390/ijerph192214622

**Published:** 2022-11-08

**Authors:** David Luna-Aleixos, Irene Llagostera-Reverter, Ximo Castelló-Benavent, Marta Aquilué-Ballarín, Gema Mecho-Montoliu, Águeda Cervera-Gasch, María Jesús Valero-Chillerón, Desirée Mena-Tudela, Laura Andreu-Pejó, Rafael Martínez-Gonzálbez, Víctor M. González-Chordá

**Affiliations:** 1Hospital Universitario de La Plana, Nursing Department, Universitat Jaume I, 12006 Castelló de la Plana, Spain; 2Nursing Research Group (GIENF Code 241), Nursing Department, Universitat Jaume I, 12006 Castelló de la Plana, Spain; 3Mathematics Department, Universitat Jaume I, 12006 Castelló de la Plana, Spain; 4Hospital Comarcal Universitario de Vinarós, Nursing Research Group (GIENF Code 241), Nursing Department, Universitat Jaume I, 12006 Castelló de la Plana, Spain; 5Hospital Universitario de La Plana, 12540 Villarreal, Spain; 6Nursing and Healthcare Research Unit (INVESTÉN-ISCIII), Institute of Health Carlos III, 28029 Madrid, Spain

**Keywords:** nurses, nursing, nursing assessment, hospitalization, validation study

## Abstract

Nursing assessment is the basis for performing interventions that match patient needs, but nurses perceive it as an administrative load. This research aims to develop and validate a meta-instrument that integrates the assessment of functional capacity, risk of pressure ulcers and risk of falling with a more parsimonious approach to nursing assessment in adult hospitalization units. Specifically, this manuscript presents the results of the development of this meta-instrument (VALENF instrument). A cross-sectional study based on recorded data was carried out in a sample of 1352 nursing assessments. Socio-demographic variables and assessments of Barthel, Braden and Downton indices at the time of admission were included. The meta-instrument’s development process includes: (i) nominal group; (ii) correlation analysis; (iii) multiple linear regressions models; (iv) reliability analysis. A seven-item solution showed a high predictive capacity with Barthel (R^2^adj = 0.938), Braden (R^2^adj = 0.926) and Downton (R^2^adj = 0.921) indices. Likewise, reliability was significant (*p* < 0.001) for Barthel (ICC = 0.969; τ-b = 0.850), Braden (ICC = 0.943; τ-b = 0.842) and Downton (ICC = 0.905; κ = 7.17) indices. VALENF instrument has an adequate predictive capacity and reliability to assess the level of functional capacity, risk of pressure injuries and risk of falls.

## 1. Introduction

Nurses working in hospitalization units are responsible for assessing, planning, implementing and re-assessing the care that patients require throughout the healthcare process, and documenting all this in their health records. Nevertheless, nurses perceive healthcare documenting as an administrative load owing to increasing quantities of data and duplicated items [1]. In the meantime, the implementation of electronic health records (HER) has prolonged data-recording times, increased workloads [2,3], cut direct healthcare times [4] and rendered nursing assessments incomplete, inconsistent and inaccurate [5].

Nursing assessments are the first step in the nursing process and can be defined as a planned, systematic, continuous and deliberate process of the collection, classification and categorization of individualized information to recognize individuals’ responses to their health problems and real or potential needs [6]. These assessments form the basis for making diagnoses and performing interventions that match patient needs [7]. Thus, any mistake or missing information, or using instruments with low validity or reliability, can affect the next steps in the nursing process and result in fragmented and incomplete care with repercussions on healthcare quality, user satisfaction and the development of adverse effects [8].

Some studies have centered on improving the workflow of nursing assessments in EHR [9,10]. Others have focused on improving contents of nursing assessment templates, mainly vital signs and physical examinations [11]. Nonetheless, nurses employ several instruments to assess the risk of developing nursing sensitive outcomes, such as a loss of functional capacity, pressure ulcers or falls. These instruments share dimensions and items that make them redundant [12]. Thus, developing a meta-instrument that integrates these instruments and takes a more parsimonious approach to nursing assessments is feasible [13].

The literature recommends arranging nursing assessments using a nursing-specific model or framework [14], such as the 14 basic needs of Henderson [15] or the 11 Functional Health Patterns of Gordon [16]. However, structured nursing assessments that are not based on a discipline-specific framework can be found [13]. It is also necessary to determine the items or variables that nurses must assess as part of their competencies, in their knowledge area and according to user profiles [17]. Depending on the nature of this information, it can be obtained by holding interviews, making observations, performing physical examinations, reviewing medical records, running diagnosis tests or applying a wide range of questionnaires to assess the risk of suffering nursing-sensitive outcomes, such as loss of functional capacity [18], pressure ulcers [19] or falls [20].

Despite their relevance, nurses perceive nursing assessments as an administrative load [1,2,4], which has increased with the implementation of EHR [3,21]. In fact, different studies have evidenced that recording nursing assessments does not meet suitable information quantity and quality standards. For instance, Paans et al. (2010) [22] found that nursing assessments had not been recorded in 20% of the health records audited in a sample of 10 Dutch hospitals. Lindo et al. (2016) [23] reported that more than 60% of the health records audited in three Jamaican hospitals did not include complete nursing assessment data. Iula et al. (2020) [24] indicated that assessments about pain and nutritional status were missing in a sample of 12,513 clinical records that were audited in an Italian hospital. Others show a failure to complete records with instruments that assess functional capacity, falls or pressure ulcers [25,26,27,28].

Some factors that might justify this situation include increased patient complexity and heavy workload [29], the variety of nursing terminologies and classifications [30], EHR developed in the traditional printed format and not considering nurses’ views [27], increased data quantity and duplicated items and the diversity of assessment instruments [1]. Palese et al. (2012) [31] concluded that nurses routinely employ between 1 and 10 assessment instruments, which can vary depending on the clinical context, units and hospitals. In another study, Redley and Raggatt (2017) [28] found that nurses in hospitalization units use between 8 and 27 assessment instruments.

The instruments employed to assess functional capacity [18], risk of pressure ulcers [19] and risk of falls [20] are probably the most widely used by nurses in adult hospitalization units. In clinical practice, these instruments are independently employed but share constructs, dimensions and items related to mobility, hygiene, eating or the elimination of body waste [11,32], which implies that items become redundant and are duplicated [1]. However, using redundant assessment instruments leads to a certain level of skepticism and a perceived waste of time, which makes it difficult for them to be accepted and implemented in nursing [10]. Therefore, nursing assessments can become an automatic and inaccurate task without much nurse engagement, which affects not only their validity, but also the task of detecting at-risk patients [31]. Consequently, we developed a research project that aims to develop and validate a meta-instrument that integrates the assessment of functional capacity, risk of pressure ulcers and risk of falling with a more parsimonious approach to nursing assessment in adult hospitalization units. Specifically, this manuscript presents the results of the development of this meta-instrument (VALENF instrument).

## 2. Materials and Methods

### 2.1. Design and Setting

A cross-sectional study based on recorded data was carried out in the Hospital de La Plana in the Valencian Community (Spain). This is the reference hospital of one health department and covers around 200,000 inhabitants, according to data from the Municipal Register.

### 2.2. Participants and Sample

The target population comprised patients over 18 years old admitted to one of the seven adult hospitalization units in the participating hospital. Special services (intensive care, emergency, operating theatres or resuscitation), home hospitalization, maternal-infant and obstetrics hospitalization units did not form part of this study due to differences in the type of care processes, in the organizational model of these units or in the assessment instruments used.

The unit of analysis was nursing assessments. Thus, the study included nursing assessments of functional capacity (Barthel index), risk of pressure ulcers (Braden index) and risk of falls (Downton scale) in the first 24 h after admission to ensure that data related to the time of admission were obtained for all patients. Otherwise, the exclusion criteria were nursing assessments of patients transferred from other units at the same hospital, or at another hospital because their assessments when hospitalized did not correspond to the initial assessment.

The literature recommends a sample size between 5 and 10 subjects per item to develop and validate assessment instruments [33]. The items for each instrument considered in the study totaled 21, which means that 210 nursing assessments was the minimum necessary sample size. However, no specific recommendations about sample size were found when combining or unifying several instruments. Notwithstanding, Palese et al. (2016) [11] used a sample with 1446 nursing assessments for a theoretical work with a similar objective. Therefore, considering that the maximum representativeness of the users of these services was sought, and as the analysis strategy required working with different subsamples, all the nursing assessments that complied with the selection criteria and were made during a four-month period (September 2021–January 2022) were included in this study.

### 2.3. Variables and Instruments

The study included socio-demographic variables (age and gender), and variables related to the healthcare process were included, such as process type (medicine, surgical), admission type (scheduled, emergency), main diagnoses according to International Classification of Diseases v-10 (ICD-10) and comorbidities (measured with Charlson index [34]). Nursing assessment-related variables were also included as scores (overall and for each item) at the time of admission, using the following instruments:Barthel index: This assesses the functional capacity (or dependency level) to carry out basic activities of daily life. It comprises 10 items, with a total score range between 0 and 100, and groups the patients into four levels (total dependency = zero–15; severe dependency = 20–35; moderate dependency = 40–55; low dependency > 60 points) [35]. González et al. (2017) [36] validated this in a Spanish population over 65 years old admitted to hospitalization units with good internal consistency (α > 0.8) and good construct validity (RMSEA < 0.08; LI > 0.9).Braden index: This assesses the risk of pressure injuries. It comprises six items with four response categories. Its scores range from six to 23 points, and it is classified into four categories (high risk = 6–12; moderate risk = 13–14; low risk = 15–18; no risk = 19–23). According to Moreno Pina et al. (2007) [37], it is considered the most appropriate instrument to assess the risk of pressure injuries in the context of the study (sensitivity = 0.27–1; specificity = 0.26–0.92; positive predictive value = 0.08–0.77, negative predictive value = 0.71–1).Downton scale: This assesses the risk of falls and comprises five items that score zero or one points. Higher scores indicate higher risk of falls, and scores above two points indicate a high risk of falls (sensitivity = 0.58; specificity = 0.62) [38].

### 2.4. Data Collection

The nurses working in the include hospitalizations units carried out data collection as part of their normal work through the EHR between September 2021 and January 2022. In February 2022, the pseudonymized database was requested from the documentation service of the participating hospital along with the variables to be studied, but without including any personal data that could identify patients. A consensus was reached beforehand with the documentation service regarding the structure of this database, and this service kept the original database with patients’ identification details.

We should note that the participating hospital has a nursing assessment protocol. This protocol indicates that nursing assessments must be made for all patients in the first 24 h after being admitted to hospitalizations units. This protocol also specifies the use of the assessment instruments considered here (Barthel Index, Braden Index and Downton Scale). These instruments are completed in the EHR, which allows for data to be exported and pseudonymized.

### 2.5. Development and Data Analysis Procedures

A descriptive analysis of the sample was performed in accordance with the nature of the variables. The existence of significant differences in the scores of Barthel index, Braden index and Downton scale were studied with the Mann Whitney U test (two groups) or Kruskal Wallis test (three or more groups) by considering the hospitalization units, as well as the process type (medical or surgical) and admission type (emergency or scheduled). Non-parametric statistics were used, since it was previously confirmed with Kolmogorov Smirnov’s test that the data did not follow a normal distribution. Moreover, the Spearman correlation tests were used for correlation analysis (negligible if ρ < 0.30; low if ρ = 0.30–0.49; moderate if ρ = 0.50–0.69; high if ρ = 0.70–0.89; very high if ρ = 0.90–1.00) [39].

After this initial analysis, the procedure to develop and validate the meta-instrument was performed by adapting the proposal by Palese et al. (2016). Specifically, this article presents the procedure followed for the development of the VALENF instrument. Firstly, a nominal group made up of five members of the research team analyzed the nursing care represented in the three instruments under the physical care dimension of the Fundamentals of Care Framework [40]. In addition, this group evaluated the similarities and conceptual redundancies between the items of the three instruments to establish direct relationships (items linked to the same care) or indirect relationships (related items linked to different care). Once this was completed, the correlations between the three instruments and relationships between items established by the nominal group were studied with the Spearman correlation test.

Next, three multiple linear regressions models were performed, one with each assessment instrument (Barthel, Braden and Downton) as dependent variables. Based on the conceptual and correlation analysis, the items of the assessment instruments and the variables Charlson index, type of process, type of hospitalization, age and sex were the independent variables. This procedure was performed with the stepwise method, adding and removing items and variables one by one to establish the most parsimonious combination of items and variables with the greatest possible predictive capacity. The Adjusted Coefficient of Determination (R^2^adj) was used as a reference to assess the predictive capacity of the models, and collinearity was studied with the Variance Inflation Factor (non-collinearity if VIF = 1–5) [41].

Based on the above results, the reliability of the original and predicted scores of the assessment instruments was analyzed. On the one hand, the agreement was studied using the one-way random for single measures Intra-class Correlation Coefficient (good agreement if ICC > 0.7) [42]. On the other hand, participants were grouped into the categories of the three assessment instruments based on the predicted scores. The concordance index (overall and for each category), Kendall’s Tau-b test (τ-b) (Barthel and Braden) and Cohen’s Kappa (κ) (Downton) with respect to the original categories were studied (good concordance if τ-b and κ > 0.7) [43]. The statistical analysis was performed with software R Commander v2.8-0 and JAMOVI v2.3.13. The significance level was stablished at *p* < 0.05.

### 2.6. Ethical Considerations

This study was accepted by the manager of the participating hospital and positively evaluated by the Ethics and Research Committee in December 2020 (code VALENF. 09/12/2020). This study complied with the Organic Law 3/2018, of 5 December, about Personal Data Protection and Guaranteeing Digital Rights, as specifically indicated by its additional 17th disposition, section d, which considers the lawful use of pseudonymized personal data for health research purposes, particularly for biomedicine. Therefore, the Ethics and Research Committee approved the request for an exemption from informed consent.

## 3. Results

### 3.1. Descriptive Analysis of the Sample

Two thousand seven hundred admissions to the adult hospitalization units were included in the study. Of these, 1040 (38.5%) nursing assessments were excluded at admission for not having performed at least one of the three assessment instruments (Barthel, Braden, Downton). Moreover, 147 (5.4%) cases were excluded because the scales were completed after the first 24 h of admission, 12 (0.5%) cases for being transfers from other units or hospitals, 65 (2.4%) cases because the instruments were incomplete and 84 (3.1%) because they belonged to another unit. Thus, the final sample consisted of 1352 (50.1%) nursing assessments.

Table 1 presents a descriptive analysis of the sample. A total of 52.1% (705) of the nursing assessments included in the study were carried out on men, and the mean age of the sample was 67.69 (±17.92; Min = 18; Max = 101) years. Four hundred and eighty-one different main diagnoses were identified, the most frequent being bronchitis not otherwise specified (ICD10 code U07.1; 10.6%; *n* = 140) and pneumonia due to unspecified microorganism (ICD10 code J18.9; 3.9%; *n* = 51). Moreover, 66.9% (*n* = 905) of the cases were medical processes, and 83.4% (*n* = 1128) were emergency admissions. The traumatology (26.6%; *n* = 359) and surgery and gynecology (20.7%; *n* = 280) units contributed almost 50% of the sample.

### 3.2. Bivariant Analysis of the Assessment Instruments

Regarding the assessment instruments, the mean score of the Barthel index was 78.38 (±33.77) points, the Braden index obtained 18.97 (±3.86) points and the Downton scale was 1.15 (±1.23) points (Table 1). Table 2 shows that there were significant differences in the mean scores of the three instruments. with almost all the variables included in the study (*p* < 0.001) except between the Braden index and gender, although this was close to significance (*p* = 0.063). Thus, the male patients included in the study and those undergoing a surgical procedure or scheduled admissions had a higher score in the Barthel index (greater functional capacity) and the Braden index (lower risk of pressure injuries), as well as a lower score on the Downton scale (lower risk of falls) (*p* < 0.001). Neurology and pulmonology patients had the lowest Barthel (lower functional capacity) and Braden (lower risk of pressure injuries) index scores, while the lowest Downton scale score was found in the general surgery unit (*p* < 0.001). Finally, the Charlson index showed significant correlations with the Barthel index (ρ = −0.499; *p* < 0.001), the Braden index (ρ = −0.521; *p* < 0.001) and the Downton scale (ρ = 0.504; *p* = <0.001). Similarly, age presented a significant correlation with the Barthel (ρ = −0.577; *p* < 0.001, Braden (ρ = −0.579; *p* < 0.001) and Downton (ρ = 0.554; *p* < 0.001) scores.

### 3.3. Development of the VALENF Instrument

Figure 1 presents a summary of the conceptual and correlation analysis. The color code reflects the grouping of the items according to the physical care of the Fundamentals of Care Framework. Likewise, the continuous lines show the direct relationships established by the nominal group, and the dotted lines represent the indirect relationships. Figure 1 also includes analysis of Spearman’s correlations (*p* < 0.001). In general, a high correlation was observed between the Barthel index and the Braden index (ρ = 0.80), although it was moderate between the Barthel index and the Downton scale (ρ = −0.64), as well as between the Downton scale and the Braden index (ρ = −0.65).

The nominal group grouped different items from the three assessment instruments into six care types using the Fundamentals of Care Framework (Personal cleansing, Toilet needs, Eat and drink, mobility, Comfort and Medication management). Of these, Mobility (understood as the ability to move and commute) was the care type that grouped the largest number of items (*n* = 7) and was present in the three assessment instruments. The Mobility (Barthel), Activity (Braden) and Walking ability (Downton) items were considered equivalent since they assess the person’s ability to move the body and move from one place to another, although their correlations ranged from weak to moderate (ρ = −0.26–0.67). It is worth mentioning that the Mobility (Braden) item assesses the person’s ability to change and control body position, so the nominal group considered this to include Transfer (Barthel), Stairs (Barthel) and Friction and shear (Braden), since they value specific activities that require this ability. In addition, the items grouped in this care type presented the greatest number of direct relationships, although the correlation between the items was highly variable (ρ = 0.26 and 0.78). Similarly, these items presented the greatest number of indirect relationships with Personal cleansing (ρ = 0.36–0.73) and Toilet needs (ρ = −0.37–0.73), in addition to an indirect relationship with the item Feeding (Barthel) (*p* = 0.66).

Personal cleansing and Toilet needs grouped three items each, but these care types were only reflected in the Barthel index. No equivalence or redundancy between items was considered. However, the items grouped in these care types also presented some indirect relationships with the item Moisture (Braden), with high correlations (ρ = 0.70–0.78), and moderate correlations with the items Sensory Perception (Braden) and Mental state (Downton) (ρ = −0.64–0.7).

The items Feeding (Barthel) and Nutrition (Braden) were grouped in the Eat and drink care type, although they were not considered equivalent, since Feeding (Barthel) assesses the ability to eat, while Nutrition (Braden) assesses the usual eating pattern. The correlation between these items was low (ρ = 0.44). In addition, one item was grouped in the Medication care type, but was not related to other items.

Finally, the nominal group did not group the items Sensory perception (Barthel), Previous falls (Downton), Mental state (Downton) or Sensory deficiency (Downton) in any care types of the Fundamentals of Care Framework. However, the Sensory perception (Braden) and Mental state (Downton) items were considered equivalent (ρ = 0.66), since both assess the person’s level of consciousness, while the Sensory deficiency (Downton) item assesses the presence of vision, hearing or limb problems.

Table 3 presents the best solution after performing several multiple linear regression models with each of the three instruments. This solution comprises seven items with a high predictive capacity regarding the global score of the Barthel index (R^2^adj = 0.938), Braden index (R^2^adj = 0.926) and Downton scale (R^2^adj = 0.921). However, it is necessary to mention that not all items were significant in the three assessment instruments. For example, the previous fall item was only significant in predicting the Downton scale score (*p* < 0.001). In addition, the medication (Downton) item was not significant in predicting the Barthel index score (*p* = 0.874), the mobility item was not significant in predicting the Downton scale (*p* = 0.876) and the sensory deficiency item (Downton) was not significant in predicting the Braden index (*p* = 0.302). In addition, the variables Charlson index, type of process, type of hospitalization and age and sex were not significant in the regressions, and the increase in R^2^adj did not exceed 0.004 points in any case, so they were not included in the final solution of seven items.

This seven-item solution showed a significant ICC and was greater than 0.9 points with the original scores of the three assessment instruments. Specifically, the ICC with the Barthel index was 0.969 (95% CI = 0.65–0.972; *p* < 0.001), although it decreased slightly with the Braden index (ICC = 0.943, 95% CI = 0.936–0.948; *p* < 0.001) and the Downton scale (ICC = 0.905; 95% CI = 0.895–0.914; *p* < 0.001).

Finally, Table 4, Table 5 and Table 6 present the concordance indexes between the original categories of each assessment instrument and the categories predicted by the seven-item solution. Specifically, the agreement between the original categories of the Barthel index and the predicted categories was significant (τ-b = 0.850; *p* < 0.001), with an overall CI of 89.6%, although this decreased to that in the intermediate categories (severe dependence CI = 25% and moderate dependence CI = 42.6%). In the same way, a significant agreement was obtained between the predicted categories and the original categories of the Braden index (CI = 83.94%; τ-b = 842; *p* < 0.001) and the Downton scale (CI = 93.71%; κ = 7.17; *p* < 0.001).

## 4. Discussion

The results of this study present a seven-item meta-instrument that can assess the level of functional capacity, the risk of pressure injuries and the risk of falls in adult hospitalization units. Specifically, the VALENF Instrument (its acronym in Spanish) is based on the analysis of the 21 items that are part of the Barthel, Braden and Downton instrument and offers a more parsimonious solution than the independent use of these instruments, with a high predictive capacity and reliability compared to the original instruments.

In a previous study, Palese et al. [13] proposed a meta-instrument for nursing assessment in adult hospitalization units, which also included an assessment of functional capacity, risk of pressure injuries and risk of falls. However, the authors used the ESAMED study database [44], only including patients older than 65 admitted to 12 medical units of different Italian hospitals. However, our study included a more heterogeneous sample, with patients over 18 years of age (age of majority in Spain) admitted for medical or surgical procedures in seven adult hospitalization units of the same hospital. This makes it difficult to compare the profile of our sample with previous studies, since there are few studies on admission assessment in adult hospitalization units, and in general, samples of older patients are used [13,36,45,46]. However, using the same instruments, our sample had a higher functional capacity and lower risk of pressure injuries at admission than the studies by Palese et al. [13] or González et al. [36], and this is consistent when considering the mean age of these samples. Another aspect to highlight is that 49.9% of the records were excluded because they did not meet the selection criteria, mainly due to inadequate completion of the assessment instruments, coinciding with previous studies [22,23]. In addition, the differences in the percentages of nursing assessments included according to the hospitalization units are noteworthy, since the hospital has a nursing assessment protocol. Perhaps these results could be explained by a high staff turnover, differences in nursing style, leadership [47] or the influence of supervisors [48], although other studies are necessary to corroborate this.

Another important difference with the study by Palese et al. [13] refers to the instruments that were used. It is true that both studies use the Barthel index [35] to assess functional capacity and the Braden index [37] to assess the risk of pressure injuries. However, Palese et al. [13] used the Conley scale [49] to assess the risk of falls, while we used the Downton scale [38]. This difference may be due to the wide variety of instruments that can be used for fall risk assessment [50] and, in turn, due to the limited availability of this type of instrument when validated in Spanish. In fact, the Downton and STRATIFY scales are the only instruments that can be used to assess the risk of falls, with some validation studies carried out in adult hospitalization units in Spain [51]. The STRATIFY scale (Area Under the Curve—AUC = 0.69; 95% CI = 0.57–0.8) has slightly better diagnostic accuracy than the Downton scale (AUC = 0.6; 95% CI = 0.48–0.72) in our context [52]. We carried out a retrospective study based on recorded data, and we did not have the possibility of choosing the instrument to assess the risk of falls, since the use of the Downton scale is protocoled in the center. However, a recent review concludes that there is no ideal instrument to assess the risk of falls, and it recommends the combined use of two instruments [53], possibly related to the risk factor detection approach of the instruments to assess the risk of falls. Thus, recent studies apply artificial intelligence to try to improve the detection of patients at risk of falls [54,55].

In addition, Palese et al. [13] also included the Blaylock Risk Assessment Screening Score (BRASS) [56] in their study. The BRASS index assesses the risk of a prolonged hospital stay or complex discharge, but this instrument was not used in our context. These differences evidence the variability in the components and instruments used in nursing assessments [17,28,31]. In fact, Palese et al. [13] start from four instruments with 42 items and propose a 20-item solution. Meanwhile, the VALENF Instrument allows for the assessment of functional capacity, risk of pressure injuries and risk of falls from seven items.

Despite these differences, the correlations between the global scores of the instruments used in both studies were moderate–high. Specifically, in our study, the correlation between functional capacity, risk of pressure injuries and risk of falls was higher than that obtained in the study by Palese et al. [13], but they did not perform a bivariate analysis with other study variables. In our case, all the variables included in the study showed significant differences (or close to significance) with respect to the mean score of the Barthel, Braden and Downton instruments. However, these variables did not improve the predictive ability of the VALENF Instrument when included in the multivariate models. Although we have not found previous studies with which to compare these results, we believe that this may justify the applicability of the VALENF Instrument in adult hospitalization units, regardless of the type of process, admission or medical specialty. In fact, 481 different medical diagnoses were obtained, and some studies suggest that the medical diagnosis is not the main determinant of care needs [57,58]. In future, the applicability of the VALENF Instrument may be tested by using groupers such as the Patient-Related Groups (DRG) [59].

To design a nursing assessment based on a conceptual model or nursing theory is a classic recommendation [14,15]. Following this recommendation, we performed a nominal group to identify the nursing care represented in the three instruments under the physical care dimension of the Fundamentals of Care Framework [40] and an analysis of correlations between the instruments. Meanwhile, Palese et al. [13] did not rely on any conceptual framework and only studied the correlations between the global scores of the instruments. Therefore, in this analysis, we identified six care types reflected in the three instruments. On the one hand, the Barthel index grouped four care types (Eat and drink; Toilet needs; Personal cleansing; Mobility). This draws attention, since the Barthel index is considered a one-dimensional instrument [36], although some studies have already shown that its construct validity could vary depending on the type of patient [60]. On the other hand, we identified Eat and drink, Comfort and Mobility care types in the Braden index; however, recent studies conclude that the items in this instrument related to mobility and activity have the greatest predictive capacity on the development of pressure injuries [61]. Finally, the Downton scale grouped Medication management and Mobility care, although there were three items that the nominal group did not identify with any of the care types included in the physical care dimension of the Fundamentals of Care Framework [40]. In addition, it is necessary to highlight the large number of items that were related to Mobility and their correlations with items related to Personal Cleansing and Toilet Needs. Thus, the VALENF Instrument included two items on Mobility, one item on Comfort, one item on Medication management and three items not related to any care type. However, it is noteworthy that the final solution did not include items related to Eat and drink or Personal Cleansing care. These care types are important for nursing, and we believe that this may indicate that there is room to improve the VALENF Instrument, including the assessment of other care types included in the Fundamentals of Care Framework, such as Eat and drink or Rest and sleep [40].

Thus, the adjusted coefficients of determination in the multivariate models were close to 0.95 with respect to the scores of the original instruments, indicating a high predictive capacity. In addition, the VALENF Instrument also showed high reliability according to the ICC results, although the agreement rates when classifying the participants were not always this high. We cannot contrast these results with previous studies, since Palese et al. [13] did not include this type of analysis in their work and based their proposal on the development of the factor analysis and structural equation models. However, it is possible that the concordance indices for the intermediate categories of the Barthel [35] and Braden [37] indices may improve if classification techniques with cluster analysis or discriminant analysis are used in future prospective studies, since there is a diversity of cut-off points and groupings for these two questionnaires in the literature [62,63]. Meanwhile, the Downton scale has limited sensitivity and specificity in our context [38], and this may affect the results of the concordance index. Thus, it is necessary to improve the assessment of the risk of falls in our context, and the VALENF Instrument could be a good starting point.

Finally, the results of this study should be considered with caution due to some limitations. Thus, we analyze the relationships between the dimensions of care and the items through a nominal group, but other consensus techniques, such as Delphi, may be more appropriate. In addition, correlation analyses and multiple regression models were used to develop the final seven-item solution, although artificial intelligence techniques could be used in future developments [54]. Finally, reliability and concordance were studied on the same sample, while applying cross-validation techniques in prospective studies would provide greater rigor to this type of instrument [64]. In addition, this is a retrospective study based on recorded data and carried out in a single hospital, which implies a possible information bias and makes it difficult to generalize the results. Despite these limitations, we want to highlight the obtained results and their interest to nurses and managers, since they represent a new approach to the design and development of nursing assessment instruments, which could speed up the nursing assessment of adult hospitalization units, improve the quality of information and reduce bureaucracy. However, it is necessary to advance in the analysis of the psychometric properties of the VALENF Instrument; therefore, in another article (part 2) we present the results related to content validity, construct validity and inter-observer reliability.

## 5. Conclusions

The VALENF Instrument (its acronym in Spanish) is a meta-instrument for nursing assessment that allows for assessments of functional capacity, risk of pressure injuries and risk of falls in hospitalization units. The VALENF Instrument was developed from a combination of the items of the Barthel, Braden and Downton indices. It is a more parsimonious, seven-item solution with a high predictive capacity and reliability compared to the original instruments. However, it is necessary to advance in the analysis of its psychometric properties and its diagnostic precision.

## Figures and Tables

**Figure 1 ijerph-19-14622-f001:**
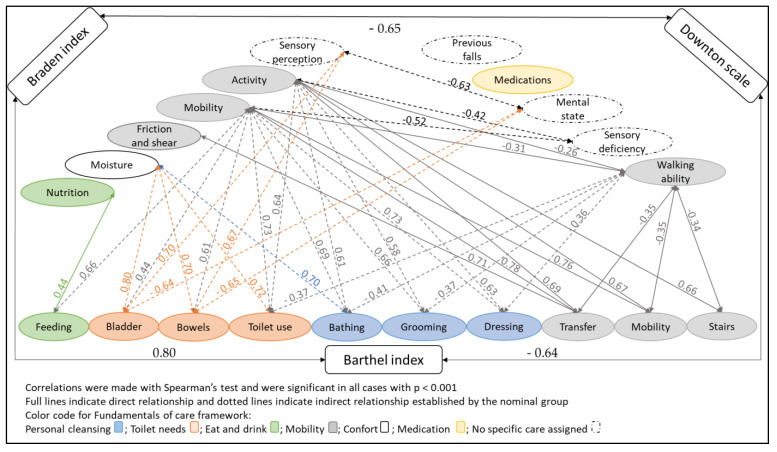
Direct and indirect relationships and Spearman’s correlation test analysis between items.

**Table 1 ijerph-19-14622-t001:** Descriptive analysis of the sample.

Variable		m (ds) ^1^
Age		67.69 (17.92)
Charlson index		3.68 (2.5)
Barthel index		78.38 (33.77)
Braden index		18.97 (3.86)
Downton scale		1.15 (1.23)
		**% (*n*) ^2^**
Sex	Male	52.1 (705)
	Female	47.9 (647)
Process type	Medical	66.9 (905)
	Surgical	33.1 (447)
Admission type	Emergency	83.4 (1128)
	Scheduled	16.6 (224)
Hospitalization unit	Traumatology	26.6 (359)
	Surgery and gynecology	20.7 (280)
	Cardio/gastroenterology	14.3 (194)
	Neuro/pulmonology	13.1 (177)
	General surgery	2.3 (31)
	Otolaryngo/urology	9 (122)
	Internal medicine	14 (189)

^1^ Mean (standard deviation); ^2^ Percentage (sample).

**Table 2 ijerph-19-14622-t002:** Bivariant analysis of nursing assessment instruments.

Variables		Barthel Index		Braden Index		Downton Scale	
		m (ds) ^1^	*p* ^2^	m (ds) ^1^	*p* ^2^	m (ds) ^1^	*p* ^2^
Sex	Male	81.83 (31.98)	<0.001 *	19.19 (3.78)	0.063	1.07 (1.21)	<0.001 *
	Female	74.62 35.28)		18.74 (3.95)		1.25 (1.24)	
Process type	Medical	73.5 (36.8)	<0.001 *	18.34 (4.18)	<0.001 *	1.3 (1.28)	<0.001 *
	Surgical	88.26 (23.76)		20.25 (2.73)		0.85 (1.05)	
Admission type	Scheduled	94.71 (16.05)	<0.001 *	21.09 (1.76)	<0.001 *	0.66 (0.863)	<0.001 *
	Emergency	75.14 (35.4)		18.55 (4.03)		1.25 (1.27)	
Hospitalization unit	Traumatology	74.33 (34.92)	<0.001 **	18.57 (4.03)	<0.001 **	1.26 (1.15)	<0.001 **
	Surgery and gynecology	82.21 (31.35)		19.52 (3.63)		0.99 (1.32)	
	Cardio/gastroenterology	77.52 (36.13)		18.32 (3.91)		1.11 (1.31)	
	Neuro/pulmonology	68.27 (37.92)		17.92 (4.03)		1.68 (1.29)	
	General surgery	82.74 (26.73)		19.54 (3.22)		0.77 (1.17)	
	Otolaryngo/urology	86.63 (26.72)		19.94 (3.41)		1.09 (1.08)	
	Internal medicine	84.68 (30.30)		19.82 (3.63)		0.82 (0.96)	

* Mann Whitney U test; ** Kruskal Wallis test; ^1^ Mean (standard deviation); ^2^
*p*-value.

**Table 3 ijerph-19-14622-t003:** Multiple linear regressions models.

Variables		Barthel Index	Braden Index	Downton Scale
Coefficient	β (IC95%) ^1^ t (*p*) ^2^ β SE ^3^	−19.839 (−16.326–−11.077) −11.077 (<0.001) 1.791	2.042 (1.602–2.482) 9.105 (<0.001) 0.224	1.632 (1.487–1.776) 22.18 (<0.001) 0.073
Barthel	β (IC95%) ^1^	4.416 (4.26–4.571)	0.095 (0.076–0.114)	−0.018 (−0.025–−0.012)
Mobility	t (*p*) ^2^	56.4 (<0.001)	9.712 (<0.001)	−5.82 (<0.001)
(VIF = 3.35) ^4^	β SE ^3^	0.08	0.009	0
Braden	β (IC95%) ^1^	4.484 (3.45–5.516)	1.149 (1.29–1.548)	−0.215 (−0.257–−0.172)
Sensory perception	t (*p*) ^2^	8.532(<0.001)	21.57 (<0.001)	−9.965 (<0.001)
(VIF = 3.09) ^4^	β SE ^3^	0.53	0.065	0
Braden	β (IC95%) ^1^	6.196 (5.27–7.12)	1.326 (1.209–1.442)	−0.136 (−0.175–−0.098)
Moisture	t (*p*) ^2^	13.07 (<0.001)	22.341 (<0.001)	−7.033 (<0.001)
(VIF = 2.69) ^4^	β SE ^3^	0.47	0.059	0
Braden	β (IC95%) ^1^	2.804 (1.83–3.778)	1.847 (1.725–1.696)	0.003 (−0.036–0.042)
Mobility	t (*p*) ^2^	5.653 (<0.001)	29.731 (<0.001)	0.156 (0.876)
(VIF = 3.19) ^4^	β SE ^3^	0.5	0.062	0
Downton	β (IC95%) ^1^	−0.816 (−2.08–0.448)	0.019 (−0.138–0.178)	1.015 (0.958–1.062)
Previous fall	t (*p*) ^2^	−1.267 (0.205)	0.244 (0.807)	38.136 (<0.001)
(VIF = 1.13) ^4^	β SE ^3^	0.64	0.081	0
Downton	β (IC95%) ^1^	−0.076 (−1.03–0.872)	−0.131 (−0.25–−0.012)	1.074 (1.035–1.113)
Medication	t (*p*) ^2^	−0.159 (0.874)	−2.174 (0.03)	54.065 (<0.001)
(VIF = 1.12) ^4^	β SE ^3^	0.48	0.061	0
Downton	β (IC95%) ^1^	−2.688 (−3.93–−1.447)	0.082 (−0.073–−0.237)	1.008 (0.957–1.0598)
Sensory deficiency	t (*p*) ^2^	−4.249 (<0.001)	1.034 (0.302)	38.8 (<0.001)
(VIF = 1.62) ^4^	β SE ^3^	0.63	0.079	0
Summarized model	R^2^ * R^2^ adjusted ANOVA (*p*)	0.939 0.938 2937 (<0.001)	0.927 0.926 2424 (<0.001)	0.922 0.921 2266 (<0.001)

^1^ Coefficient and confidence index at 95%; ^2^
*t*-test and *p*-value; ^3^ standard error of the coefficient; ^4^ Variance Inflation Factor; * Coefficient of Determination.

**Table 4 ijerph-19-14622-t004:** Concordance Index between original and predicted Barthel index categories.

Predicted Categories		Original Categories				
		Total	Severe	Moderate	Slight	Total
Total	*n*	151 (86.3%) *	12	0	0	163
Severe	*n*	21	9 (25%) *	12	10	52
Moderate	*n*	3	15	23 (42.6%) *	49	90
Slight	*n*	0	0	19	1028 (94.6%) *	1047
Total	*n*	175	36	54	1087	1352 (89.6%) *

* Between parentheses, the overall concordance index and for each category can be consulted.

**Table 5 ijerph-19-14622-t005:** Concordance Index between original and predicted Braden index categories.

Predicted Categories		Original Categories				
		High	Moderate	Low	No Risk	Total
High	*n*	68 (70.8%) *	31	0	9	99
Moderate	*n*	26	64 (58.7%) *	32	0	122
Low	*n*	2	14	185 (70.3%) *	66	267
No risk	*n*	0	0	46	818 (92.5%) *	864
Total	*n*	96	109	263	884	1352 (83.94%) *

* Between parentheses, the overall concordance index and for each category can be consulted.

**Table 6 ijerph-19-14622-t006:** Concordance Index between original and predicted Downton scale categories.

Predicted Categories		Original Categories		
		No Risk	Risk	Total
No risk	*n*	1139 (99.7%) *	82	1221
Risk	*n*	3	128 (61%) *	131
Total	*n*	1142	210	1352 (93.71%) *

* Between parentheses, the overall concordance index and for each category can be consulted.

## Data Availability

Data are available upon reasonable request. All necessary data are supplied and available in the manuscript; however, the corresponding author will provide the dataset upon request. All data relevant to the study are included in the article.

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
