# Peer review of "Development and Validation of a Meta-Instrument for Nursing Assessment in Adult Hospitalization Units (VALENF Instrument) (Part I)"

_ijerph, 2022, doi:10.3390/ijerph192214622_

Round 1
Reviewer 1 Report
The meta-instrument (VALENF instrument- 110 mind). developed by the authors from traditional instruments would facilitate nursing surveillance work in adult units. The adaptation process is adequate and the article presents the information very neatly. Validation in different contexts is expected.
Author Response
Dear review,
We sincerely would like to appreciate the comments and contributions made on our work.
On the following pages, you will find these comments where a brief answer of the comment made is included and the line where the change was made, when necessary.
Sincerely

Reviewer 2 Report
Thank you for submitting your manuscript.
In the Design and Setting section, there are 200,000 people treated at the hospital, adds the period.
Specify the exclusion criteria in Methodology.
What are the specific characteristics of the population to which the instrument is administered?
I think that in your study the COVID data of the sample is not necessary, delete it.
Line 195-196, change the word "manuscript".
Limitations:
I do not think there is a lack of agreed methodology to develop instruments based on others. There is an exictra methodology, but you haven't used it in this study (in part 1, at least not). For this reason, I do not believe that the lack of consensus is a limitation, it is a limitation not to have used the methodology.
There is a time bias. You have to do a prospective study to properly assess reliability.
If you want this instrument to be valid and reliable in the future, you have to analyze its psychometric properties and also its measurement properties, including structural validation, responsiveness, comprehensibility analysis and readability of the final instrument.
I hope that the changes and proposals will be considered.
Thank you
Author Response

(The authors gave the same response as above.)

Reviewer 3 Report
This manuscript is an interesting contribution that helps to clarify a still hazy and poorly explored area of nursing assessment. However, there are some inaccuracies listed below:
1) The authors chose the Downton Scale to assess the risk of falls; however, this scale (according to a 2021 systematic review) has been extensively analyzed in contexts that do not include acute patients such as nursing homes, while when applied to the hospital environment, its ability to predict falls significantly decreases. Continued at point 4).
2) Line 165: The Downton Scale does not assess the risk of pressure injury, but the risk of falls.
3) Lines 373-379: Although the authors included patients over 18 years of age, the mean age of the sample was 67.69 (±17.92) years (as indicated in the lines 245-246); please clarify this in the paragraph. (NB: In the Palese study the population was between 73 and 75 years old, in the present study between 50 and 85 years old.)
4) Lines 399-401: The Downton scale is not the only one validated in Spanish; according to the 2021 systematic review, the STRATIFY scale was also validated in Spanish in 2017. According to this systematic review “The STRATIFY scale is a predictive tool for the risk of falls in hospitalized patients. The compilation of the scale is not performed through direct observation of the patient, but the evaluator reports the score based on information obtained from the previous observation or from other caregivers. STRATIFY has been extensively studied in intensive care units in Australia, Europe and Canada and has also been applied in numerous geriatric and rehabilitation departments. In these contexts, it has long been considered the “Gold standard” tool to be used at patient admission thanks to the high sensitivity value demonstrated by numerous studies (between 73.7% and 93.0%) and the simplicity and speed of application (3 min).”
As in point 1), the authors are invited to explain clearly and with evidence-based arguments the reasons for which the Downton Scale was chosen and not other scales. To solve points 1) and 4), I recommend a thorough reading of the following papers:
- Aranda-Gallardo M, Enriquez de Luna-Rodriguez M, Vazquez-Blanco MJ, Canca-Sanchez JC, Moya-Suarez AB, Morales-Asencio JM. Diagnostic validity of the STRATIFY and Downton instruments for evaluating the risk of falls by hospitalised acute-care patients: a multicentre longitudinal study. BMC Health Serv Res. 2017 Apr 17;17(1):277. doi: 10.1186/s12913-017-2214-3. PMID: 28412939; PMCID: PMC5393002.
- Strini V, Schiavolin R, Prendin A. Fall Risk Assessment Scales: A Systematic Literature Review. Nurs Rep. 2021 Jun 2;11(2):430-443. doi: 10.3390/nursrep11020041. PMID: 34968219; PMCID: PMC8608097.
5) Lines 467-468: The authors claim to have carried out a conceptual analysis through a nominal group; however, it is not clear which methodology was used. Carrying out a conceptual analysis requires a rigorous methodology (for example, Rodgers' evolutionary concept analysis or Walker & Avant's approach to concept analysis can be used). Authors are asked to clarify which technique was used. If no conceptual analysis method has been used, authors are requested to remove any periphrases that state that a conceptual analysis has been performed, from every part of text and tables (including abstract).
I remain at your disposal for further clarifications.
Best regards.
Author Response

(The authors gave the same response as above.)

Round 2
Reviewer 3 Report
the manuscript has been sufficiently improved to warrant publication in IJERPH.